# Unraveling the Osteogenic Activity and Molecular Mechanism of an Antioxidant Collagen Peptide in MC3T3-E1 Cells

**DOI:** 10.3390/nu17050824

**Published:** 2025-02-27

**Authors:** Yali Wang, Yue Wang, Xiaoyan Zhuang, Yonghui Zhang, Baishan Fang, Yousi Fu

**Affiliations:** 1Department of Chemical and Biochemical Engineering, College of Chemistry and Chemical Engineering, Xiamen University, Xiamen 361005, China; wangyal1@msu.edu; 2Department of Biochemistry & Molecular Biology, Michigan State University, East Lansing, MI 48824, USA; 3Department of Joint Surgery and Sports Medicine, Zhongshan Hospital of Xiamen University, Xiamen 361005, China; xmwangyue1992@gmail.com; 4College of Food and Biological Engineering, Jimei University, Xiamen 361021, China; zxy@jmu.edu.cn (X.Z.); yhz@jmu.edu.cn (Y.Z.); 5The Key Laboratory for Synthetic Biotechnology of Xiamen City, Xiamen University, Xiamen 361005, China

**Keywords:** antioxidant peptide, osteoporosis, integrin, molecular docking, Akt, collagen I, MC3T3-E1 osteoblasts

## Abstract

**Background**: Osteoporosis has become an inevitable health issue with global aging, and the current drug treatments often have adverse side effects, highlighting the need for safer and more effective therapies. Collagen-derived peptides are promising alternatives due to their favorable safety profile and biological activity. This study aimed to investigate the osteogenic and anti-apoptotic properties of collagen peptide UU1 (GASGPMGPR) in addition to its antioxidant activity. **Methods**: The effects of UU1 were evaluated in MC3T3-E1 cells by assessing osteogenic markers, including alkaline phosphatase (ALP), Cyclin D1, runt-related transcription factor 2 (Runx2), and Akt/β-catenin signaling. Western blot analysis quantified collagen I, osteocalcin, and phosphorylated Akt levels. Anti-apoptotic effects were measured via *p*-Akt levels and the Bax/Bcl-2 ratio. Computational molecular docking was performed to explore the molecular mechanism of UU1 via its interaction with epidermal growth factor receptor (EGFR) and collagen-binding integrin. **Results**: UU1 treatment promoted cell differentiation, with elevated ALP, Cyclin D1, Runx2, and Akt/β-catenin signaling. Notably, at 0.025 mg/mL, UU1 upregulated the levels of collagen I, osteocalcin, and phosphorylated Akt by 2.14, 3.37, and 1.95 times, respectively, compared to the control. Additionally, UU1 exhibited anti-apoptotic effects, indicated by increased *p*-Akt levels and a reduced Bax/Bcl-2 ratio. Molecular docking analysis suggested that UU1 could assist the dimerization of EGFR, facilitating downstream signaling transductions and activating collagen-binding integrin. **Conclusions**: These findings highlight UU1 as a multifunctional peptide with antioxidant, osteogenic, and anti-apoptotic properties, positioning it as a promising candidate for anti-osteoporosis applications in the food and pharmaceutical industries.

## 1. Introduction

Osteoporosis is a common chronic disease characterized by significant bone loss and decreased bone strength, currently affecting about 200 million people worldwide and causing over 8.9 million fractures annually [1]. The incidence of osteoporosis increases with aging, and by 2050, the number of people over the age of 60 will rise to 2.1 billion worldwide, which suggests that osteoporosis is becoming an increasingly critical public health issue [2]. Bone remodeling homeostasis is maintained by osteoblasts (bone-forming cells) and osteoclasts (bone-resorbing cells); when this balance is disrupted, progressive bone loss and structural weakening occur, leading to osteoporosis. Stimulating the activity of osteoblasts to accelerate new bone formation is an approach to prevent osteoporosis [3]. Pharmaceutical approaches for osteoporosis, while effective in increasing bone density, may lead to significant long-term side effects, such as atypical fractures or cardiovascular issues [4]. As an alternative, oral supplements containing food-derived active collagen peptides have gained attention for their ability to safely and effectively support bone health without these adverse effects [5].

As safe nutraceuticals, collagen supplements come from different sources (e.g., porcine, bovine, marine) and in diverse types (e.g., peptides, hydrolysate, gelatin) [6]. Several studies have reported taking collagen hydrolysate or small peptides as oral supplements for pressure ulcers, xerosis, skin aging, anti-aging, and wound healing [7,8,9]. Collagen peptides are viewed as a complementary approach alongside established osteoporosis treatments, such as calcium and vitamin D [10]. Specific collagen peptides have been shown to improve bone mineral density in postmenopausal women with age-related osteoporosis [11], and various peptide hydrolysates derived from livestock, peptides from cattle, and porcine bone collagen have been reported to enhance osteoblastic activity in MC3T3-E1 cells [12,13]. Despite the promise of collagen peptides, challenges remain. The quality, bioavailability, and effectiveness of collagen hydrolysate supplements vary widely, and active peptides from collagen blends are often neither isolated nor thoroughly elucidated. Low-molecular-weight peptides generally offer higher bioavailability and biological activity compared to larger peptides and parent proteins owing to their ease of absorption, stabilization, and enhanced interaction with biological targets [14]. This highlights the need to identify and characterize highly efficacious collagen peptides with low molecular weights.

In our preliminary work, we isolated and identified low-molecular-weight collagen peptides from yak (*Bos grunniens*), an animal mainly indigenous to the Qinghai-Tibet Plateau and traditionally regarded as a source of health-promoting food [15]. Some of these peptides exhibited antioxidant properties in vitro and improved the lifespan and oxidative stress resistance in nematodes [15,16]. Antioxidant peptides typically exhibit a range of biological activities, including anti-inflammatory, immunomodulatory, and anti-osteoporosis functions, and are promising candidates for functional foods designed to address various health conditions. This suggests that our active antioxidant collagen peptides have great research and economic values and are worthy of further study for more precise functions, such as their role in bone metabolism regulation.

In this research, we aimed to investigate the osteogenic potential of a novel antioxidant collagen peptide, UU1 (GASGPMGPR), on MC3T3-E1 cells, a widely used model for studying bone growth and formation. Key factors involved in osteoblast proliferation, differentiation, and mineralization—such as alkaline phosphatase (ALP), runt-related transcription factor 2 (Runx2), collagen I, osteocalcin, β-catenin, and Akt signaling—were evaluated, along with apoptosis-related proteins from the BCL-2 family. Furthermore, computational molecular docking studies were performed to explore the interaction between UU1 and key osteogenic receptors, including the epidermal growth factor receptor (EGFR) and integrins, offering insights into the potential molecular mechanisms underlying UU1’s osteogenic activity. In summary, the functional peptide UU1 represents a compelling candidate for anti-osteoporosis applications, particularly in the development of functional foods and nutraceuticals for the aging population. This research not only addresses a critical gap in the identification of bioactive, low-molecular-weight collagen peptides but also offers novel insights into their potential therapeutic use in osteoporosis prevention and treatment.

## 2. Materials and Methods

### 2.1. Chemicals and Reagents

The peptide UU1 was synthesized by GenScript Biotech Co., Ltd. (Nanjing, China) with a purity of no less than 98%. Fetal bovine serum (FBS) was bought from Gibco (Carlsbad, CA, USA). Minimum essential medium alpha (α-MEM) was purchased from HyClone (Beijing, China). Penicillin-streptomycin and 0.25% trypsin-EDTA, as well as phosphate-buffered saline (PBS), were purchased from Solarbio Science & Technology Co., Ltd. (Beijing, China). The BCA assay kit and MTT assay kit were from Solarbio Science & Technology Co., Ltd. (Beijing, China). RIPA lysis buffer (P0013K), alizarin red S, ascorbic acid, β-glycerophosphate, and bovine serum albumin (BSA) were purchased from Beyotime Biotechnology (Shanghai, China). Anti-Runx2 (ab236639), anti-Collagen I (ab255809), anti-Osteocalcin (ab133612), anti-Akt1 + Akt2 + Akt3 (phospho S472 + S473 + S474) (ab192623) and anti-Akt1 + Akt2 + Akt3 (ab200195), anti-beta Catenin (ab6301), anti-Cyclin D1 (ab40754), anti-Bax (ab182734), and anti-Bcl-2 (ab16904) antibodies were purchased from Abcam (Toronto, ON, Canada). Anti-β-actin antibody was obtained from Sangon Biotech (Shanghai, China). The other chemicals and reagents were of analytical grade.

### 2.2. Cell Culture

The mouse pre-osteoblast cell line MC3T3-E1 was purchased from American Type Culture Collection (ATCC, Manassas, VA, USA). The cells were cultured in α-MEM medium with 10% FBS and 1% penicillin-streptomycin and incubated at 37 °C in a humidified atmosphere of 95% air and 5% CO_2_. At about 80% confluence, the cells were subcultured into new sterile cell culture plates for further experiments after 0.25% trypsin-EDTA treatment. Unless otherwise noted, the cells (1 × 10^4^ cells per well) were incubated in 6-well plates for 24 h and then treated with 0.0125, 0.025, and 0.05 mg/mL UU1 for specific durations in different tests. The treatment groups were designated as U-12.5, U-25, and U-50, and the cells treated without UU1 were set as the control group.

### 2.3. Cell Viability Assay

Cell viability assay was determined by the MTT assay kit. The instructions in the MTT assay kit were followed. Briefly after 0.25% trypsin-EDTA treatment, the cells (1 × 10^4^ cells per well) were seeded in 96-well plates for 24 h and then treated with UU1 (0.0125, 0.025, and 0.05 mg/mL) for another 24 h. The old medium was replaced with 90 μL of fresh medium (α-MEM medium, 10% FBS, 1% penicillin-streptomycin) and 10 μL of tetrazolium dye (MTT) solution for 4 h. Then the supernatant was carefully discarded, 110 μL of formazan solvent was added to each well, and the plates were gently shaken at low speeds for 10 min; after that, the absorbance was measured at 490 nm by a plate reader (Infinite 200 PRO, TECAN, Männedorf, Switzerland). The blank group contained only the medium, MTT, and formazan mixture solution. All groups were performed in triplicate or more. The cell viability was then calculated using the following formula: The cell viability (%) = [(Abs. of sample − Abs. of blank)/(Abs. of control − Abs. of blank)] × 100%.

### 2.4. Alkaline Phosphatase Assay

The activity of ALP was determined using an ALP assay kit from Beyotime Biotechnology (Shanghai, China) according to the instruction protocol. Briefly, the cells were grown in 6-well culture plates and then treated with UU1 (0.0125, 0.025, and 0.05 mg/mL) for 7 days. After treatment, the cells were harvested and washed twice with PBS. Subsequently, they were lysed with RIPA lysis buffer and centrifuged at 4 °C at 12,000× *g* for 5 min. The protein concentration of the supernatant was measured using a BCA assay kit. The ALP activity of the supernatant was determined by incubating it with *p*-nitrophenol (10 mM) for 10 min at 37 °C and then stopping the reaction. The absorbance of the reaction solution was read at 405 nm using a plate reader. The standard curve of *p*-nitrophenol was plotted, and net ALP activity was calculated.

### 2.5. Mineralization

The degree of mineralization was determined by the Alizarin red S staining kit from Solarbio (Beijing, China). The medium used for the mineralization study consisted of ascorbic acid (50 mg/mL), dexamethasone (10 nM), and β-sodium glycerophosphate (2 mM) in α-MEM (10% FBS, 1% penicillin-streptomycin). The cells were treated with different concentrations of UU1 (0.0125, 0.025, and 0.05 mg/mL) and incubated in 24-well plates for 15 days. Following the instructions, the medium was removed, and the cells were washed once with PBS. Subsequently, the cells were fixed for 20 min and then washed thrice with PBS. They were then stained with Alizarin S red for 30 min at 25 °C. After staining, the cells were washed with sufficient Milli-Q water (Millipore, Burlington, MA, USA) and observed under white light. After dissolving the stain in the plate with 10% cetylpyridinium chloride (*w*/*v*), the absorbance was observed at 562 nm using a plate reader.

### 2.6. RNA Isolation and Real-Time Quantitative PCR

The cells were seeded in 6-well plates with α-MEM containing 10% FBS and 1% penicillin–streptomycin for 24 h; afterward, the cells were treated with 0.0125, 0.025, and 0.05 mg/mL UU1 for 7 days. The total RNA was extracted from the cells using the Total RNA Isolation Kit from Vazyme Biotech (Nanjing, China). Complementary DNA (cDNA) was synthesized by HiScript III All-in-one RT SuperMix Perfect for qPCR (Vazyme, Nanjing, China) according to the manufacturer’s protocol. The samples for RT-qPCR were prepared using the SYBR qPCR Master Mix from Vazyme Biotech (Nanjing, China). The genes and their primers are listed in Table 1. The quantified results were calculated using the 2^−ΔΔCt^ method.

### 2.7. Western Blotting

The cells were treated with peptide UU1 (0.0125, 0.025, and 0.05 mg/mL) for 7 days, as described before. After that, the cells were harvested, washed, and then lysed in RIPA lysis buffer with protease and phosphatase inhibitors. The prepared sample (40 μg) was electrophoresed in 15% SDS-PAGE, transferred to a polyvinylidene fluoride (PVDF) membrane, and blocked with 5% nonfat dried milk and 1% BSA in TBST. Membranes were incubated with corresponding antibodies overnight at 4 °C. Subsequently, the secondary antibody was incubated at room temperature for 50 min. The protein bands were imaged by ChemiDoc Touch (BIO-RAD, Hercules, CA, USA). The results of band assays were expressed as the fold change relative to the corresponding untreated control.

### 2.8. Molecular Docking Analysis

The 3D structures of the receptors were obtained from the Protein Data Bank (PDB): EGFR bound to epidermal growth factor (PDB ID: 3NPJ) [17], the EGFR ligand-binding head bound to TGF-α (PDB ID: 1MOX) [18], and the integrin α2 I domain (PDB ID: 1DZI) [19]. These receptors and peptide ligand UU1 were prepared by AutoDockTools (version 1.5.7) prior to docking. The original binding sites in the crystal structure were selected as docking sites. Molecular docking was performed on AutoDock Vina [20]. Each docking was conducted at least 10 times. The 2D interactions of receptor–ligand complexes were generated by LigPlot+ (version 2.2.5) [21]. The molecular structures were visualized by PyMOL 3.7. During docking, the ligand UU1 molecule was designated as flexible with 25 active torsions, while all receptors were set as rigid.

### 2.9. Statistical Analysis

All data results are presented as the mean ± standard deviation in this study. All tests were duplicated at least three times. One-way analysis of variance (ANOVA) followed by Dunnett’s test was utilized in the data analysis using R (version 4.3.3); *p* < 0.05 was considered statistically significant.

## 3. Results and Discussion

### 3.1. Peptide UU1 Enhanced MC3T3-E1 Cell Proliferation, Differentiation, and Mineralization

The effect of different concentrations of peptide UU1 (0.0125, 0.025, 0.05 mg/mL) treatment on the proliferation activity of MC3T3-E1 cells was assessed by MTT assay. As shown in Figure 1a, in the U-12.5 and U-25 groups, treated with 0.0125 and 0.025 mg/mL UU1, respectively, cell viability showed an increasing trend compared to the control group, though these changes were not statistically significant (*p* > 0.05). In the U-50 group, which was treated with 0.05 mg/mL UU1, there was a significant increase in cell proliferation compared to the control group (*p* < 0.05), with a 45.11% increase in viability. These results reveal that peptide UU1 treatment could effectively enhance the proliferation of MC3T3-E1 cells, with the 0.05 mg/mL UU1 treatment demonstrating the most pronounced effect.

In the process of osteogenic differentiation, osteoblasts produce extracellular matrix and accumulate mineral deposition by secreting serial factors to facilitate mineralization. Alkaline phosphatase (ALP) is a glycoprotein that plays a key role in osteoblast differentiation and mineralization. Increased ALP expression or activity accelerates differentiation by supplying abundant phosphate, which is essential for mineral deposition [22]. Therefore, ALP has been considered a classical biochemical marker for evaluating osteoporosis and osteoblast activity [23]. To assess the impact of peptide UU1 on osteoblast differentiation, the mRNA expression and enzymatic activity of ALP were measured in MC3T3-E1 cells after 7 days of treatment with varying concentrations. As shown in Figure 1b,c, peptide UU1 treatment led to an increase in ALP mRNA expression level and activity. In the U-12.5, U-25, and U-50 groups, the ALP mRNA expression levels were 1.23, 1.39, and 1.12 times higher than that of the control group, respectively, although these increases were not statistically significant (*p* > 0.05). In terms of enzymatic activity, ALP activity increased by 28.70% in the U-12.5 group, 53.40% in the U-25 group (*p* < 0.05), and 7.10% in the U-50 group compared to the control group. These results indicate that peptide UU1 had a positive influence on osteoblast differentiation, particularly at a concentration of 0.025 mg/mL, which showed the most significant increase in ALP activity.

To further assess the effect of peptide UU1 on osteogenic differentiation in MC3T3-E1 cells, we analyzed the protein expression of key relevant factors, including Runx2, Collagen I, and Osteocalcin by Western blot after 7 days of treatment. Runx2 is a master transcription factor critical for bone development and is involved in multiple processes in osteogenesis, including differentiation, matrix production, and mineralization [24,25]. Overexpression of Runx2 resulted in enhanced cell differentiation, while declined Runx2 resulted in the maturational arrest of osteoblasts [26,27]. Furthermore, several osteoblast-specific matrix protein genes are controlled by Runx2, such as ALP, osteocalcin, and collagen I, of which collagen I accounts for about 90% of bone [28]. As shown in Figure 1d,e, the Runx2 protein levels in the U-125, U-25, and U-50 groups were significantly elevated compared to the control group, with increases of 1.28-, 1.31-, and 1.21-fold, respectively (*p* < 0.01). Type I collagen levels were also markedly stimulated by UU1 treatment, with a distinct dose-dependent trend. In the U-25 and U-50 groups, the collagen I levels were 2.14- and 2.53-fold higher than in the control group, respectively (*p* < 0.001). Additionally, during bone formation, osteocalcin is a key essential factor whose expression occurs later than that of Runx2 [24]. The osteocalcin levels were significantly upregulated in all UU1-treated groups compared to the control (*p* < 0.01). Notably, the U-25 group exhibited the highest osteocalcin level, which was 3.37 times that of the control (*p* < 0.0001).

Finally, we assessed the effect of peptide UU1 on the mineralization of MC3T3-E1 cells. Mineralization is a hallmark of the final stage of osteoblast maturation, crucial for bone matrix formation [29,30]. Representative photographs of the mineralized cells, stained with Alizarin Red S after 15 days of treatment, are presented in Figure 1f,g. Apparently, the mineralization of MC3T3-E1 cells under UU1 treatments was significantly enhanced compared to the control. In the U-12.5, U-25, and U-50 groups, the corresponding mineralized nodule content of MC3T3-E1 cells increased by 26.27%, 21.54%, and 21.68%, respectively (*p* < 0.001), indicating a clear effect of UU1 in promoting mineral deposition.

The studies above show that peptide UU1 not only increased both the activity and mRNA expression of ALP but also impressively elevated the expression levels of Runx2 and Collagen I after 7 days of treatment. These results suggest that peptide UU1 could promote osteoblast differentiation from the early stage and eventually facilitate the osteoblast’s maturity. Further evidence to support the osteoblastic activity of peptide UU1 was provided by the increased secretion of osteocalcin, a secreted factor affecting matrix mineralization. Osteocalcin is typically expressed 4–5 days later than Runx2 during mouse development [31,32]. Moreover, mineralization, one of the signs of maturity for osteoblasts, was markedly enhanced after 15 days of UU1 treatment. Overall, these results suggest that the collagen-derived antioxidant peptide UU1 exerted a stimulatory effect on the osteogenesis process in MC3T3-E1 cells, particularly at the higher concentrations, by promoting early differentiation and advancing osteoblast maturation. It is notable that osteoporosis involves the dysregulation of both bone formation and resorption. Future studies will explore peptide UU1’s impact on osteoclast activity, as well as its role in modulating osteoblast–osteoclast communication.

### 3.2. Peptide UU1 Activated β-Catenin and Akt Signaling in MC3T3-E1 Cells

Osteogenesis is a complex, multi-week process involving several signaling networks, such as the Wnt/β-catenin, Akt, MAPK, and TGF-β pathways [33]. Among these, the factors of β-catenin and Akt signaling are fundamental in regulating the proliferation and differentiation in MC3T3-E1 cells. Many lines of evidence have shown that β-catenin plays a crucial role in the formation of stable cell adhesion structures and interacts with cell adhesion molecules [34]. Activation of the Wnt/β-catenin pathway has been shown to promote osteogenic differentiation, and it is through this pathway that the osteogenic activity of *Euodia sutchuenensis* Dode extract, geraniin, and liraglutide is mediated [35,36]. Furthermore, the hydrolysates complex from porcine bone and yak bone were demonstrated to promote the proliferation and differentiation in osteoblasts through PI3K/Akt and Wnt pathways [37,38].

The effect of peptide UU1 on the expression of Akt and β-catenin was further investigated. The Akt mRNA expression was upregulated with increasing content of UU1 (Figure 2a, *p* < 0.001). Additionally, the gene transcription levels of β-catenin were significantly increased to 5.39, 7.76, and 3.63 times in the U-12.5, U-25, and U-50 groups, respectively, compared to the control (Figure 2b, *p* < 0.01). Afterward, the protein expression levels of β-catenin, Akt, and phosphorylated Akt (*p*-Akt) were analyzed by Western blot (Figure 2c–e). Consistent with the pattern observed in mRNA expression, the protein expression of β-catenin in the U-25 group was significantly increased, which was 1.56 times that of the control (Figure 2d, *p* < 0.001). Regarding the *p*-Akt/Akt ratio, in the U-25 group, it increased to 1.95 times that of the control (Figure 2e, *p* < 0.0001), indicating that phosphorylation of Akt is more pronounced by peptide UU1 treatment. There were reports that phosphorylation of Akt not only exerts an anti-apoptotic effect but also regulates various cellular processes, including cell survival, growth, and metabolism [39]. These results revealed that UU1 treatment up-regulated the expression of total Akt and β-catenin at both gene and protein levels and significantly accelerated the phosphorylation of Akt.

### 3.3. The Regulation of Peptide UU1 on Cell Cycle and Apoptosis

In the early stage of osteogenesis, cell proliferation is essential which supplies sufficient osteoblasts for bone recruitment. Moreover, the decrease in the proliferative activity of osteoblasts has been associated with the development of osteoporosis [40]. Proper proliferation and differentiation are proven to be tightly dependent on the support of the cell cycle [41]. Enhancing the progression of the cell cycle can provide abundant cells to accelerate the formation of new bone. The cell division kinases are also known as cyclin-dependent kinases involved in the control of the cell cycle, while the cyclins play as regulators [42]. Cyclin D1 is a positive regulator through the G1 phase in the cell cycle progression [43]. As the doses of peptide UU1 treatment increased, the protein level of Cyclin D1 correspondingly increased, in which the Cyclin D1 level of the U-50 group was 1.51 times that of the control (Figure 3a,b, *p* <0.05).

Next, we evaluated the potential anti-apoptotic effect of peptide UU1 on MC3T3-E1 cells, retarding the apoptosis in osteoblasts could support the synthesis of mineral matrix and reduce osteoclast-mediated bone resorption, thereby protecting the long-term bone health from osteoporosis [44]. The BCL-2 family of proteins, including pro-apoptotic Bax and anti-apoptotic Bcl-2, comprises a group of proteins for manipulating apoptosis [45]. Bax induces structural changes in the mitochondrial outer membrane to promote permeabilization, while Bcl-2 inhibits Bax to prolong cell survival and delay apoptosis, which implies that the Bax/Bcl-2 ratio affects osteoblasts and osteoclast apoptosis, with a higher ratio reducing cellular resistance to apoptotic stimuli [46]. As shown in Figure 3a,c, with the increase of peptide UU1 treatment, the Bax/Bcl-2 expression ratio was notably attenuated (*p* < 0.01). The lowest Bax/Bcl-2 ratio was observed in the U-50 group, which was significantly decreased to approximately 59% of the control (Figure 3c, *p* < 0.001).

Taken together, peptide UU1 was observed to have multiple beneficial effects, including promoting cell proliferation, enhancing osteoblast maturation, and reducing apoptosis. These findings suggest that peptide UU1 may have potential as a bioactive molecule in osteoporosis. Our future research will explore peptide UU1’s anti-apoptotic and anti-osteoporotic activities in mammalian models to gain deeper insight into the molecular mechanism by which peptide UU1 may affect osteoblasts during osteogenesis. In addition, we plan to investigate the long-term effects of peptide UU1 treatment, including its ability to sustain osteogenic activity over extended periods.

### 3.4. Computational Molecular Docking

To investigate the molecular mechanisms by which peptide UU1 influences osteoblast function during osteogenesis, we performed molecular docking studies of peptide UU1 with key osteogenic receptors, including the epidermal growth factor receptor (EGFR) and integrin. EGFR is a receptor tyrosine kinase whose extracellular module conformation has been revealed differently binding to epidermal growth factor (EGF) transforming growth factor α (TGF-α), epigen, epiregulin, etc. As a potential receptor for bovine-derived peptide binding, EGFR has received extensive attention in molecular docking and screening [37]. Upon ligand-induced dimerization, the receptor tyrosine kinase is activated to initiate intracellular signaling pathways regulating various processes, including proliferation, differentiation, and apoptosis. Crystal structures of 3NJP and 1MOX consist of a truncated human EGFR bound to EGF and TGF-α, respectively. The peptide UU1 was able to dock into the cleave and bind to EGFR and EGF/TGF-α simultaneously. As shown in Figure 4a,b, in 3NJP, Gln8, Gly9, Ser11, Gly317, and Asp344 of EGFR and Gln43 of EGF formed hydrogen bonds with peptide UU1. In the UU1-1MOX complex, residue His45 of TGF-α and Gln8, Ser11, Asn40, Tyr64, Gln408, and His409 of EGFR were involved in forming hydrogen bonds with peptide UU1 (Figure 4c,d). The binding energies of the two docking complexes were similar, namely −7.4 kcal/mol for UU1-3NPJ and −7.3 kcal/mol for UU1-1MOX. Such a subtle difference may be due to the fact that when EGFR is bound to TGF-α, the pocket is just slightly more compressed compared to its conformation when bound to EGF [47]. Comparing two docking conformations, the O atoms of Ala 2 and Pro 5, the guanidine group of Arg 9, and Ser 3 in peptide UU1 were revealed to be the common active binding hotspots. Based on the results, peptide UU1 enhanced the interaction between EGFR and its ligands, leading us to hypothesize that it facilitates ligand-induced dimerization of EGFR. This, in turn, may more easily activate multiple intracellular signaling pathways involved in cell growth.

Integrins are a family of heterodimeric cell adhesion proteins consisting of non-covalently associated α and β subunits, which interact with extracellular matrix (ECM) components, transmit bidirectional signals across the cell membrane, and mediate adhesion, cell growth, and differentiation. Four of twenty-four known integrins are categorized as collagen-binding integrins. The collagen receptor integrin α subunits are characterized by an inserted I domain as the major recognition site for ECM ligands. Several structures of integrin α1I and α2I domains in complex with mimic collagen peptides, such as GFOGER and GLOGEN, have been reported [48]. Since we found that peptide UU1 greatly enhanced the expression of collagen I and cell differentiation in MC3T3-E1 cells, collagen-binding integrin α2β1 came into our sight, which prefers binding collagen I and thus is a very likely binding target for UU1. The docking results of 1DZI integrin α2 I domain and peptide UU1 are shown in Figure 5. Peptide UU1 bound across the upper edge of the α2 I domain with −5.8 kcal/mol binding affinity. Most residues of UU1 were engaged, forming a complementary surface with 1DZI (Figure 5a). The side chains of Ala2 and Pro8 in UU1 faced down and buried into the α2 I domain, making hydrophobic contact with the receptors. As shown in Figure 5b, eight H-bonding interactions with integrin were from the backbone and Arg9 in peptide UU1. Notably, Arg9 of UU1 plugged into the acidic pocket of Glu256, Ser257, Glu299, and Gly255, contributing to the majority of polar interactions. There are findings imply that collagen-integrin receptor interaction is a crucial signal for the osteoblastic differentiation of bone [49]. Furthermore, GPR and GPRP- have been reported to interact with αxβ2 integrin. Since peptide UU1 contains a GPR motif, αxβ2 is another potential binding target in vivo [50]. Other integrins, such as α1β1 and α11β1, which relate to bound to different types of collagen [51], might also be affected by peptide UU1. More studies are needed to reveal them. Taken together, we found that when the UU1-integrin complex was initially formed, it became possible for it to start playing a signaling role of activating cell proliferation and differentiation in the early stage of osteoblasts. Under exogenous stimulation, collagen I may be induced and accumulate quickly, and then more collagen-integrins will follow, providing an even more positive stimulus for cell growth.

## 4. Conclusions

Overall, this study revealed that the antioxidant peptide UU1 demonstrated significant potential as a therapeutic agent for osteoporosis, enhancing the cell cycle and promoting cell proliferation, resulting in an earlier onset of differentiation, and ultimately accelerating cell maturity in MC3T3-E1 cells. In addition, UU1 exhibited an anti-apoptotic effect by reducing the Bax/Bcl-2 ratio and stimulating *p*-Akt. Molecular docking analysis revealed the capacity of UU1 binding to EGFR and integrin α2 through which it may mediate various pathways in cell growth and collagen assembly. UU1, as a safe and antioxidant bioactive yak bone-derived peptide, held great potential for functional products to improve bone health, providing a valuable foundation for the development of novel therapeutic strategies for osteoporosis.

## Figures and Tables

**Figure 1 nutrients-17-00824-f001:**
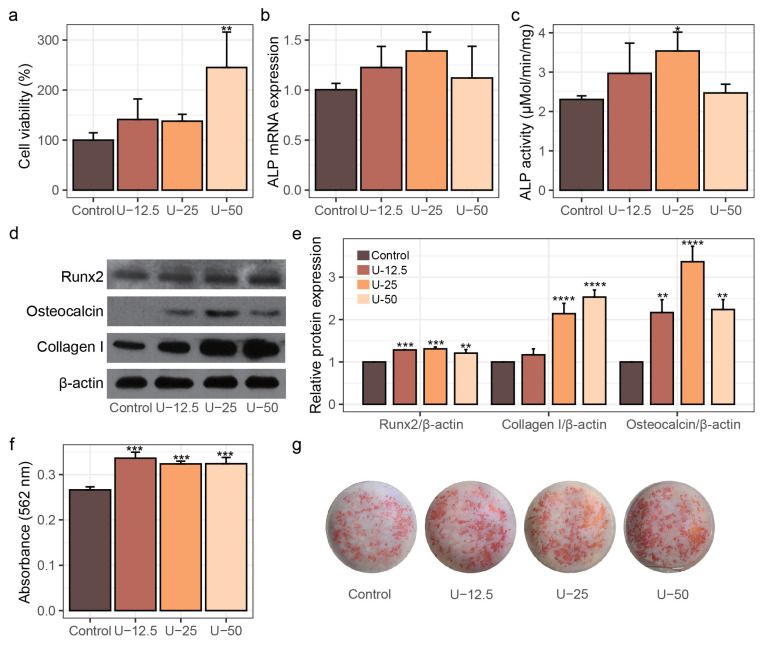
The effect of peptide UU1 (0.0125, 0.025, 0.05 mg/mL) treatment on cell viability, differentiation, and mineralization in MC3T3-E1 cells. (**a**) The proliferation activity of MC3T3-E1 cells detected by MTT assays. (**b**) The ALP mRNA expression level and (**c**) activity of ALP. (**d**) Western blot of the expression of Runx2, Collagen I, Osteocalcin, and (**e**) their relative expression. (**f**) The mineralized deposit after 15 days of treatment was measured at 562 nm. (**g**) The representative images of mineralized nodes stained with Alizarin Red S. The protein expression data are represented as means ± standard deviation. Data were analyzed using one-way ANOVA with Dunnett’s test, * *p* < 0.05, ** *p* < 0.01, *** *p* < 0.001, and **** *p* < 0.0001 indicate significant differences compared to the control group.

**Figure 2 nutrients-17-00824-f002:**
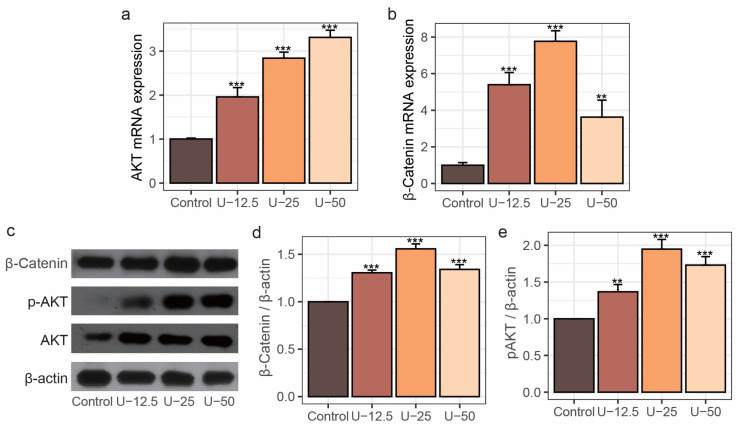
The stimulative effect of UU1 (0.0125, 0.025, 0.05 mg/mL) treatment on β-catenin and Akt signaling in MC3T3-E1 cells. (**a**,**b**) Total Akt and β-catenin mRNA expression levels. (**c**) Western blot of the expression of β-catenin, *p*-Akt, Akt, and (**d**) β-catenin protein and (**e**) *p*-Akt/Akt ratio. Data are represented as means ± standard deviation and analyzed using one-way ANOVA with Dunnett’s test, and ** *p* < 0.01 and *** *p* < 0.001 indicate significant differences compared to the control group.

**Figure 3 nutrients-17-00824-f003:**
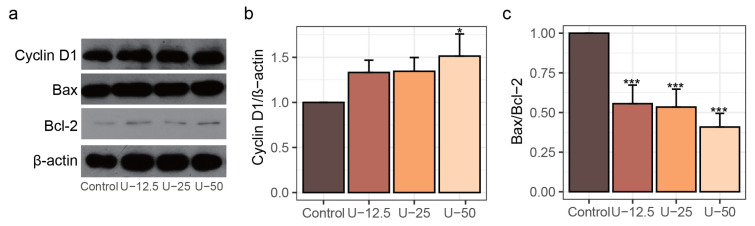
The implications of UU1 (0.0125, 0.025, 0.05 mg/mL) treatment on Cyclin D1, Bax, and Bcl-2 in MC3T3-E1 cells. (**a**) Western blots of Cyclin D1, Bax, and Bcl-2 proteins. The relative expression level of (**b**) Cyclin D1 and (**c**) Bax/Bcl-2. Data are represented as means ± standard deviation and analyzed using one-way ANOVA with Dunnett’s test, and * *p* < 0.05 and *** *p* < 0.001 indicate significant differences compared to the control group.

**Figure 4 nutrients-17-00824-f004:**
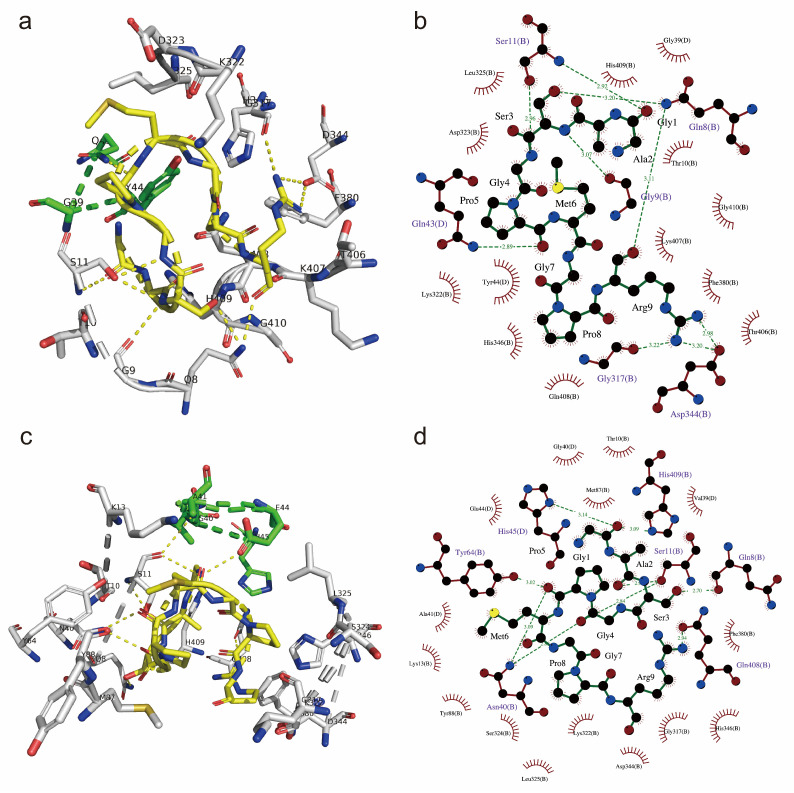
Molecular docking of peptide UU1 and epidermal growth factor receptor EGFR complex (PDB ID:3NJP and 1MOX). (**a**) the conformation and (**b**) the 2D diagram of interactions in UU1-3NJP. (**c**) the conformation and (**d**) the 2D diagram of interactions in UU1-1MOX. Displayed as sticks in (**a**,**c**); EGFR (chain B) is white, EGF/TGF-α (chain D) is green, and UU1 is yellow. The green dashed line in 2D diagrams (**b**,**d**) plotted by LigPlot+ indicated the hydrogen bond and its length. Eyelash-shaped red items indicate the non-ligand residues involved in hydrophobic contact(s).

**Figure 5 nutrients-17-00824-f005:**
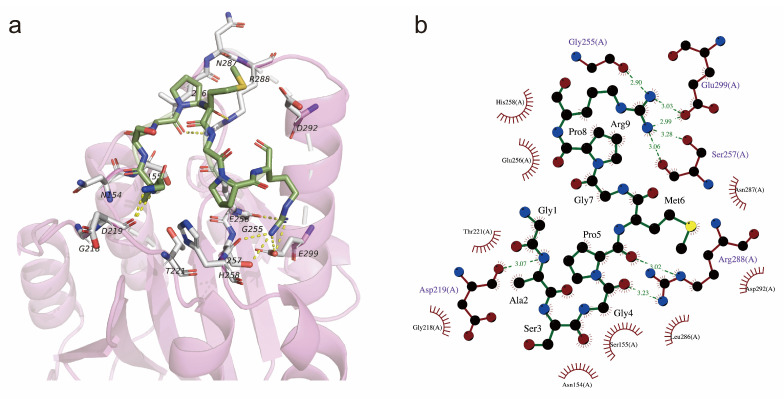
The molecular docking conformation of UU1 and integrin α2 I domain. (**a**) The structure of UU1 and 1DZI binding complex. 1DZI was exhibited translucently in red cartoon mode, residues interacting with UU1 are shown in white stick mode, and UU1 is in green stick mode. (**b**) Two-dimensional plot of interactions in UU1-1DZI; the items in the plot were as described above.

**Table 1 nutrients-17-00824-t001:** Primers for RT-qPCR.

Gene	Forward (5′ → 3′)	Reverse (5′ → 3′)
ALP	CCAACTCTTTTGTGCCAGAGA	GGCTACATTGGTGTTGAGCTTTT
Akt	ATGAACGACGTAGCCATTGTG	TTGTAGCCAATAAAGGTGCCAT
β-Catenin	TCATCATTCTGGCCAGTG	AGAGCAGACAGACAGCACT
β-Actin	GGCTGTATTCCCCTCCATCG	CCAGTTGGTAACAATGCCATGT

## Data Availability

The original contributions presented in the study are included in the article, further inquiries can be directed to the corresponding authors.

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
