# Peer review of "Unraveling the Osteogenic Activity and Molecular Mechanism of an Antioxidant Collagen Peptide in MC3T3-E1 Cells"

_nutrients, 2025, doi:10.3390/nu17050824_

Round 1

Reviewer 1 Report

Comments and Suggestions for Authors

This study explores the osteogenic and anti-apoptotic properties of the antioxidant collagen peptide UU1 (GASGPMGPR) in MC3T3-E1 cells, emphasizing its potential as a therapeutic agent for osteoporosis. UU1 promotes cell proliferation, differentiation, and mineralization, upregulates key osteogenic markers, and demonstrates anti-apoptotic effects through Akt/β-catenin signaling. However, the statistical analysis requires clarification to support the findings, and the in vitro nature of the results limits speculation. The authors should address how UU1 might combat osteoporosis, considering the communication between osteoblasts and osteoclasts (BSP), not just the mineralization process. Additionally, the paper does not explore the bone genetic regulatory network, leaving a gap in the mechanistic understanding of the subject.

Major Comments

Bone Genetic Regulatory Network (GRN) Activation: The results should be interpreted in the context of bone GRN activation following a standard sequence. Refer to the paper by Chekroun et al. and Sun et al., which discusses bone GRN activation through the β-catenin pathway.

Chekroun A et al. Theoretical evidence of osteoblast self-inhibition after activation of the genetic regulatory network controlling mineralization. J Theor Biol. 2022 Mar 21;537:111005. doi: 10.1016/j.jtbi.2022.111005. Epub 2022 Jan 12. PMID: 35031309.

M. Sun et al. Effects of matrix stiffness on the morphology, adhesion, proliferation, and osteogenic differentiation of mesenchymal stem cells. International journal of medical sciences, 15(3):257 268, 2018.

The optimal concentration of UU1 for therapeutic use is not established. A detailed analysis of the dose-response relationship and a discussion of its implications for potential therapeutic dosages are needed. Analyzing the bone GRN sequence could help predict and identify plateau effects.

The authors should clarify how this can fight osteoporosis since they are not addressing the communication between osteoblast and osteoclast  (BSP) but only the mineralization process. Please clarify 

The statistical methods are briefly mentioned and need elaboration. Clearly state the number of samples per group tested, use the Shapiro-Wilk test to assess distribution, and apply parametric or non-parametric tests accordingly. Provide a detailed explanation and justification of the statistical analyses used.

The long-term effects of UU1 treatment are not addressed. Include data or discussions about the sustained impact of UU1.

Please include a section discussing plans or preliminary results from animal models to improve the discussion.

Author Response

This study explores the osteogenic and anti-apoptotic properties of the antioxidant collagen peptide UU1 (GASGPMGPR) in MC3T3-E1 cells, emphasizing its potential as a therapeutic agent for osteoporosis. UU1 promotes cell proliferation, differentiation, and mineralization, upregulates key osteogenic markers, and demonstrates anti-apoptotic effects through Akt/β-catenin signaling. However, the statistical analysis requires clarification to support the findings, and the in vitro nature of the results limits speculation. The authors should address how UU1 might combat osteoporosis, considering the communication between osteoblasts and osteoclasts (BSP), not just the mineralization process. Additionally, the paper does not explore the bone genetic regulatory network, leaving a gap in the mechanistic understanding of the subject.

Your thorough review has been incredibly beneficial, and I am thankful for the time and expertise you dedicated to providing such detailed and constructive feedback.

Major Comments

Bone Genetic Regulatory Network (GRN) Activation: The results should be interpreted in the context of bone GRN activation following a standard sequence. Refer to the paper by Chekroun et al. and Sun et al., which discusses bone GRN activation through the β-catenin pathway.

Chekroun A et al. Theoretical evidence of osteoblast self-inhibition after activation of the genetic regulatory network controlling mineralization. J Theor Biol. 2022 Mar 21;537:111005. doi: 10.1016/j.jtbi.2022.111005. Epub 2022 Jan 12. PMID: 35031309.

  1. Sun et al. Effects of matrix stiffness on the morphology, adhesion, proliferation, and osteogenic differentiation of mesenchymal stem cells. International journal of medical sciences, 15(3):257 268, 2018.

Thank you for your insightful comment. In the revised manuscript, we have added citations to Chekroun et al. and Sun et al., referencing their works.

The optimal concentration of UU1 for therapeutic use is not established. A detailed analysis of the dose-response relationship and a discussion of its implications for potential therapeutic dosages are needed. Analyzing the bone GRN sequence could help predict and identify plateau effects.

We sincerely thank the reviewer for this insightful comment. We agree that establishing the optimal therapeutic concentration of peptide UU1 is critical for translational applications. Our current data suggest a concentration-dependent trend, we recognize that systematic in vitro and in vivo dose-ranging experiments will be essential to define the optimal dosage for clinical translation. We quite agree with the reviewer's suggestion that analyzing bone GRN sequences is very valuable.

The authors should clarify how this can fight osteoporosis since they are not addressing the communication between osteoblast and osteoclast (BSP) but only the mineralization process. Please clarify

We sincerely thank the reviewer for raising this critical point. While our current study focused on the osteogenic potential of peptide UU1 in promoting osteoblast differentiation and mineralization, we acknowledge that bone remodeling involves a dynamic balance between osteoblast-mediated bone formation and osteoclast-mediated resorption. While osteoclast was not directly measured in our study, we have added a discussion highlighting this limitation and emphasizing the need for future investigations into peptide UU1’s effects on osteoclast activity and osteoblast-osteoclast cross-talk.  Please see the revised manuscript.

Lines 268-271 “It is notable that osteoporosis involves dysregulation of both bone formation and resorption. Future studies will explore peptide UU1’s impact on osteoclast activity, as well as its role in modulating osteoblast-osteoclast communication.”.

The statistical methods are briefly mentioned and need elaboration. Clearly state the number of samples per group tested, use the Shapiro-Wilk test to assess distribution, and apply parametric or non-parametric tests accordingly. Provide a detailed explanation and justification of the statistical analyses used.

We thank the reviewer for the valuable comment. We have used the Shapiro-Wilk test in R to verify the normal distribution of our data. Statistical analyses were performed in this using R. Data are presented as mean ± standard deviation (SD), as indicated. The number of samples per group was three. A p-value of < 0.05 was considered statistically significant.

The long-term effects of UU1 treatment are not addressed. Include data or discussions about the sustained impact of UU1.

We thank the reviewer for their valuable comment. We have added a discussion of the plans for future studies to investigate the long-term effects of peptide UU1 treatment.

Lines 348-350 “In addition, we plan to investigate the long-term effects of peptide UU1 treatment, in-cluding its ability to sustain osteogenic activity over extended periods.”.

Please include a section discussing plans or preliminary results from animal models to improve the discussion.

We thank the reviewer for the suggestion. In the revised manuscript, we have added sentences outlining our plans for future studies using animal models to further investigate the osteogenic effects of peptide UU1. Please see the revised manuscript.

Lines 346-348 “Our future research will explore peptide UU1’s anti-apoptotic and anti-osteoporotic activities in mammalian models to gain deeper insight into the molecular mechanism by which peptide UU1 may affect osteoblasts during osteogenesis.”.

Reviewer 2 Report

Comments and Suggestions for Authors

Osteoporosis is a progressive metabolic bone disease that is associated with a loss of bone mass due to an imbalance between bone formation (osteoblasts) and bone resorption (osteoclasts). In the present study, Wang et al. investigated the osteogenic and anti-apoptotic effects of collagen derived from Tibetan yak bones on mouse preosteoblasts (MC3T3-E1 cells) in vitro. The low molecular-weight-collagen peptide UU1 stimulates the expression of various osteogenic markers such as collagen 1, osteocalcin, Runx2 and alkaline phosphatase, indicating that the MC3T3-E1 cells have differentiated into osteoblasts. In addition, UU1 protects the cells against apoptosis and activates collagen-binding integrin. The authors suggest that the low molecular-weight-collagen peptide UU1 could be a promising food supplement for patients suffering osteoporosis.

This is a well presented paper dealing with the possible role of the yak collagen peptide UU1 in the treatment of osteoporosis. I have only a few minor comments that should be carefully considered by the authors.

Minor comments

Line (L) 125: It should read “instructions” in MMT assay….

L 209-211: This sentence is not clearly written and should be reworded.

L 256: What do the other mean with “overall metabolism?”

L 319-327: These sentences are not clearly written and should be reworded.

L 344-345: Rephrase this sentence. I don't understand what the authors are trying to say.

Fig. 2: The labeling of Figures 2d and 2e does not correspond to the legend and the text (see also L 291 and 293).

Comments on the Quality of English Language

Some sentences are not clearly written and should be reworded. See my comments.

Author Response

Osteoporosis is a progressive metabolic bone disease that is associated with a loss of bone mass due to an imbalance between bone formation (osteoblasts) and bone resorption (osteoclasts). In the present study, Wang et al. investigated the osteogenic and anti-apoptotic effects of collagen derived from Tibetan yak bones on mouse preosteoblasts (MC3T3-E1 cells) in vitro. The low molecular-weight-collagen peptide UU1 stimulates the expression of various osteogenic markers such as collagen 1, osteocalcin, Runx2 and alkaline phosphatase, indicating that the MC3T3-E1 cells have differentiated into osteoblasts. In addition, UU1 protects the cells against apoptosis and activates collagen-binding integrin. The authors suggest that the low molecular-weight-collagen peptide UU1 could be a promising food supplement for patients suffering osteoporosis.

This is a well presented paper dealing with the possible role of the yak collagen peptide UU1 in the treatment of osteoporosis. I have only a few minor comments that should be carefully considered by the authors.

We deeply appreciate the time and effort you took to provide such a careful and constructive review. 

Minor comments

Line (L) 125: It should read “instructions” in MMT assay….

This sentence has been revised. Please see the revised manuscript.

Lines 128-129 “The instructions in the MTT assay kit were followed. Briefly,”

L 209-211: This sentence is not clearly written and should be reworded.

We thank the reviewer for the comment. Please see the revised manuscript.

Lines 215-217 “Alkaline phosphatase (ALP) is a glycoprotein that plays a key role in osteoblast differ-entiation and mineralization. Increased ALP expression or activity accelerates differentiation by supplying abundant phosphate, which is essential for mineral deposition.”.

L 256: What do the other mean with “overall metabolism?”

Thank you for pointing this out. We agree that this phrase was too vague and have removed it from the text to improve clarity and precision.

L 319-327: These sentences are not clearly written and should be reworded.

We thank the reviewer for the comment. Please see the revised manuscript.

Lines 330-338 “Next, we evaluated the potential anti-apoptotic effect of peptide UU1 on MC3T3-E1 cells, retarding the apoptosis in osteoblasts could support the synthesis of mineral matrix and reduce osteoclast-mediated bone resorption, thereby protect the long-term bone health from osteoporosis [42]. The BCL-2 family of proteins, including pro-apoptotic Bax and anti-apoptotic Bcl-2, which comprises a group of proteins for manipulating apoptosis [43]. Bax induces structural changes in the mitochondrial outer membrane to promote permeabilization, while Bcl-2 inhibits Bax to prolong cell survival and delay apoptosis, which implies that the Bax/Bcl-2 ratio affects osteoblasts and osteoclast apoptosis, with a higher ratio reducing cellular resistance to apoptotic stimuli.”.

L 344-345: Rephrase this sentence. I don't understand what the authors are trying to say.

We thank the reviewer for the comment. We have rephrased the sentence to clarify its meaning and ensure grammatical accuracy.

Lines 359-362 “To investigate the molecular mechanisms by which peptide UU1 influences osteo-blast function during osteogenesis, we performed molecular docking studies of peptide UU1 with key osteogenic receptors, including the epidermal growth factor receptor (EGFR) and integrin.”.

Fig. 2: The labeling of Figures 2d and 2e does not correspond to the legend and the text (see also L 291 and 293).

We thank the reviewer for pointing this out. We have carefully reviewed the figures, legends, and text, and corrected the labeling to ensure consistency across all sections.

Reviewer 3 Report

Comments and Suggestions for Authors

1.39- Osteoporosis is a common chronic disease...
2. 45-46 When this balance is disrupted, progressive bone loss and structural weakening occur, leading to osteoporosis.
3. 48-49 "Cancer" as a side effect of osteoporosis treatment is very general and unsupported in the text by any reference to a specific mechanism or study.
4. 60-61 "Various peptide hydrolysates derived from livestock, peptides from pea protein, and desalted duck egg white have been reported to enhance osteoblastic activity in MC3T3-E1 cells."
Peptides from pea protein and egg white are not collagenous, so comparing them to collagen hydrolysates is misleading.
5. 68-69 "This highlights the need to identify and characterize highly-efficacy collagen peptides with low molecular weight."
There should be: highly efficacious collagen peptides.
6. 77-78 "More precise functions" sounds vague – what specific functions are considered?
7. 85-87 EGFR is not a typical osteogenic target – why?
8. 124-125 The instructions in the MTT assay kit were followed. Briefly,...
9. 169–178 treated with
10. 171-172 washed and then lysed
11. 176 does this refer to the expression of a specific protein?
12. 183-184 why are these sites biologically significant for UU1
13. 188-189 Strictly treating receptors as rigid may affect results, especially if natural protein movements are relevant for binding – methodological error
14. 192 analysis of variance (ANOVA).
15. 193 p < 0.05 was considered statistically significant
16. 200-201 If the changes are not statistically significant, presenting percentage differences may be misleading – it is worth emphasizing that these are trends.
17. 214-215 led to an increase in ALP mRNA expression and activity
18. 226-227 Runx2 is a master transcription factor critical for bone development and is involved in multiple processes...
19. 234-236 If the effect is dose-dependent, was it statistically significant at different levels?
20. 256-258 a sign of osteoblast maturity - this is not the only marker of osteoblast maturity
21. 259-261 "Potent" suggests a strong effect, but there is no direct comparison with other osteogenic factors
22. 275-276 β-catenin and Akt signaling are fundamental in regulating..
23. 278-280 has been shown
24. 286-287 were significantly increased
25. 288-290 the pattern observed in mRNA expression
26. 292-293 more pronounced
27. 297-298 There is no evidence that UU1 accelerates maturation, only that it affects protein expression.
28. 308-310 has been associated with the development of osteoporosis
29. 311-312 enhancing the progression of the cell cycle
30. 317 1.51 times that of the control
31. 320-321 support the synthesis of… ;reduce osteoclast-mediated bone resorption…
32. 324-325 prolong cell survival and delay apoptosis
33. 329 Incorrect sentence construction
34. 332-333 Too strong a claim
35. 344 To gain deeper insight into the molecular mechanism by which peptide UU1 may affect osteoblasts during osteogenesis.
36. 353-355 formed hydrogen bonds with UU1
37. 359-360 slightly more compressed compared to its conformation when bound to EGF
38. 363-365 Syntactic and logical errors
39. Imprecise formulations requiring clarification.
40. Incorrect grammar (e.g., verb tenses, sentence structure).
41. Too strong claims without sufficient evidence (especially regarding osteogenesis and potential osteoporosis treatment).
42. Problems with sentence construction affecting clarity. Imprecise and sometimes ungrammatical language.
43. Just because UU1 increases collagen I expression, it does not automatically mean it directly binds integrin α2β1. Other mechanisms are possible, such as indirect effects through other factors regulating collagen expression.
44. "The main collagen peptide UU1" suggests that UU1 is collagen, which is inconsistent with the earlier description. Additionally, the description of the spatial arrangement of the binding ("front upper to the slightly right edge") is imprecise and unclear.
45. Lack of evidence for the presented mechanism. It has not been demonstrated that integrin α2β1 itself stimulates further collagen I accumulation. A reference to the literature or experiments confirming such an effect would be necessary.
46. It was probably meant to be EGFR instead of EFPR.
47. The statement "safe and antioxidant bioactive yak bone-derived peptide" is not supported by data. No results were presented regarding the safety of UU1 or its antioxidant activity.
48. Simplified and illogical reasoning regarding the mechanisms of UU1.
49. Lack of references to the literature or experimental evidence in some places.

Comments on the Quality of English Language

english should be improved

Author Response

We would like to express our sincere gratitude for your detailed and conscientious review. Your careful attention to detail has significantly contributed to the improvement of our manuscript.

  1. 39- Osteoporosis is a common chronic disease...

Changes have been made in the revised manuscript.

  1. 45-46 When this balance is disrupted, progressive bone loss and structural weakening occur, leading to osteoporosis.

The sentence has been revised.

  1. 48-49 "Cancer" as a side effect of osteoporosis treatment is very general and unsupported in the text by any reference to a specific mechanism or study.

We thank the reviewer for pointing out this issue. As suggested, we have removed the word 'cancer' from the text.

  1. 60-61 "Various peptide hydrolysates derived from livestock, peptides from pea protein, and desalted duck egg white have been reported to enhance osteoblastic activity in MC3T3-E1 cells." Peptides from pea protein and egg white are not collagenous, so comparing them to collagen hydrolysates is misleading.

We thank the reviewer for the insightful comment. To address this, we have updated the reference and replaced the mention of collagenous peptides with non-collagenous peptides, ensuring the text now accurately reflects the relevant studies. Please see the revised manuscript.

Lines 62-63 “peptides from cattle and porcine bone collagen have been reported to enhance osteoblastic activity in MC3T3-E1 cells.”

  1. 68-69 "This highlights the need to identify and characterize highly-efficacy collagen peptides with low molecular weight." There should be: highly efficacious collagen peptides.

We thank the reviewer for the valuable comment. As suggested, we have revised the sentence to ‘highly efficacious’ to ensure accuracy and clarity.

  1. 77-78 "More precise functions" sounds vague – what specific functions are considered?

We thank the reviewer for the insightful comment. The updated text now highlights specific functions, such as bone metabolism regulation.

  1. 85-87 EGFR is not a typical osteogenic target – why?

We thank the reviewer for the valuable comment regarding EGFR. We acknowledge that EGFR is not a typical osteogenic target, but it is more commonly associated with cell proliferation and cancer biology. In this study, we included EGFR in the molecular docking analysis to explore potential cross-talk between signaling pathways that might indirectly influence osteogenic activity, given its role in cell growth and survival.

  1. 124-125 The instructions in the MTT assay kit were followed. Briefly,...

This sentence has been revised. Please see the revised manuscript.

Lines 128-129 “The instructions in the MTT assay kit were followed. Briefly,”

  1. 169–178 treated with

Already changed.

  1. 171-172 washed and then lysed

Line 177 “After that, the cells were harvested, washed, and then lysed in RIPA”.

  1. 176 does this refer to the expression of a specific protein?

We thank the reviewer for the question. The mentioned process does not refer to the expression of a specific protein but rather to cell lysis. We have clarified this point in the revised manuscript to avoid any confusion.

  1. 183-184 why are these sites biologically significant for UU1

We thank the reviewer for this question. The receptors were chosen based on their potential roles in cellular processes that may influence osteogenic activity, such as integrins, particularly integrin α2, are well-known for their role in cell-matrix interactions and osteoblast adhesion, which are critical for bone formation. By investigating these receptors, we aimed to explore mechanisms through which peptide UU1 might exert its osteogenic effects.

  1. 188-189 Strictly treating receptors as rigid may affect results, especially if natural protein movements are relevant for binding – methodological error

We thank the reviewer for the insightful comment. We acknowledge there is limitations of rigid receptor docking. We set the peptide as flexible, the method we used is semi-flexible docking, which is a widely used and acceptable method, especially in this case, the structure receptors are all from PDB bank, rather than predicted structures, and their binding pocket has already been revealed. We also believe there is space to improve the method in the future, dynamic molecular simulation is a good way to study the dynamic interactions between the peptide and receptors.

  1. 192 analysis of variance (ANOVA).

It has been revised.

  1. 193 p < 0.05 was considered statistically significant

The sentence has been revised.

  1. 200-201 If the changes are not statistically significant, presenting percentage differences may be misleading – it is worth emphasizing that these are trends.

We thank the reviewer for the valuable comment. We have revised the text to clarify it and please see the revised manuscript.

Line 206 “cell viability showed an increasing trend compared to the control group”.

  1. 214-215 led to an increase in ALP mRNA expression and activity

The sentence has been revised.

  1. 226-227 Runx2 is a master transcription factor critical for bone development and is involved in multiple processes...

The sentence has been revised.

  1. 234-236 If the effect is dose-dependent, was it statistically significant at different levels?

We thank the reviewer for the insightful comment. For Runx2 expression, no significant differences were observed between the U-12.5, U-25, and U-50 groups (T-Test, p > 0.05). Type I collagen expression showed statistically significant differences between the U-12.5 and U-25 groups (T-Test, p < 0.01) and between the U-12.5 and U-50 groups (T-Test, p < 0.001). However, the difference between the U-25 and U-50 groups was not significant (T-Test, p > 0.05).

  1. 256-258 a sign of osteoblast maturity - this is not the only marker of osteoblast maturity

The sentence has been revised.

Line 264 “mineralization, one of the signs of maturity for osteoblasts”.

  1. 259-261 "Potent" suggests a strong effect, but there is no direct comparison with other osteogenic factors

We thank the reviewer for the insightful comment. We have revised the text and please see the revised manuscript.

Line 266 “peptide UU1 exerted a stimulatory effect”.

  1. 275-276 β-catenin and Akt signaling are fundamental in regulating.

The sentence has been revised according to your suggestion.

  1. 278-280 has been shown

It has been revised.

  1. 286-287 were significantly increased

The sentence has been revised according to your suggestion.

  1. 288-290 the pattern observed in mRNA expression

The sentence has been revised according to your suggestion.

  1. 292-293 more pronounced

It has been revised.

  1. 297-298 There is no evidence that UU1 accelerates maturation, only that it affects protein expression.

We thank the reviewer for the insightful comment. We have revised this sentence according to your suggestion.

  1. 308-310 has been associated with the development of osteoporosis

We have revised this sentence according to your suggestion.

  1. 311-312 enhancing the progression of the cell cycle

We have revised this sentence according to your suggestion.

  1. 317 1.51 times that of the control

It has been revised.

  1. 320-321 support the synthesis of… ;reduce osteoclast-mediated bone resorption…

We have revised this sentence according to your suggestion.

  1. 324-325 prolong cell survival and delay apoptosis

We have revised this sentence according to your suggestion.

  1. 329 Incorrect sentence construction

We have revised this sentence according to your suggestion. The revised conclusion emphasizes that the results suggest the potential for further investigation rather than making definitive claims.

Lines 340-343 “The lowest Bax/Bcl-2 ratio was observed in the U-50 group, which was significantly de-creased to approximately 59% of the control (Figure 3c, p < 0.001)”.

  1. 332-333 Too strong a claim

We thank the reviewer for the comment. We have revised this sentence according to your suggestion.

Lines 343-346 “Taken together, peptide UU1 was observed to have multiple beneficial effects, including promoting cell proliferation, enhancing osteoblast maturation, and reducing apoptosis. These findings suggest that peptide UU1 may have potential as a bioactive molecule in osteoporosis.”.

  1. 344 To gain deeper insight into the molecular mechanism by which peptide UU1 may affect osteoblasts during osteogenesis.

We have revised this sentence according to your suggestion.

Lines 346-350 “Our future research will explore peptide UU1’s anti-apoptotic and anti-osteoporotic activities in mammalian models to gain deeper insight into the molecular mechanism by which peptide UU1 may affect osteoblasts during osteogenesis.”

  1. 353-355 formed hydrogen bonds with UU1

We have revised this sentence according to your suggestion.

  1. 359-360 slightly more compressed compared to its conformation when bound to EGF

We have revised this sentence according to your suggestion.

  1. 363-365 Syntactic and logical errors

Thank you for pointing this out. We have revised this sentence.

Lines 380-383 “Based on the results, peptide UU1 enhanced the interaction between EGFR and its ligands, leading us to hypothesize that it facilitates ligand-induced dimerization of EGFR. This, in turn, may more easily activate multiple intracellular signaling pathways involved in cell growth.”

  1. Imprecise formulations requiring clarification.

We have carefully reviewed the text and revised the relevant sections to provide clearer and more precise explanations.

  1. Incorrect grammar (e.g., verb tenses, sentence structure).

We have carefully reviewed the text and corrected errors in verb tenses, sentence structure, and overall grammar to improve clarity and readability.

  1. Too strong claims without sufficient evidence (especially regarding osteogenesis and potential osteoporosis treatment).

We thank the reviewer for the comment. We have revised this sentence according to your suggestion. Please see the comment 34.

  1. Problems with sentence construction affecting clarity. Imprecise and sometimes ungrammatical language.

We have carefully reviewed the text and revised the relevant sections to provide clearer and more precise explanations.

  1. Just because UU1 increases collagen I expression, it does not automatically mean it directly binds integrin α2β1. Other mechanisms are possible, such as indirect effects through other factors regulating collagen expression.

We appreciate the reviewer’s insightful comment. We acknowledge that increased collagen I expression does not necessarily indicate direct binding of peptide UU1 to integrin α2β1, and other mechanisms may be involved. Our results just suggested a potential interaction.

  1. "The main collagen peptide UU1" suggests that UU1 is collagen, which is inconsistent with the earlier description. Additionally, the description of the spatial arrangement of the binding ("front upper to the slightly right edge") is imprecise and unclear.

Thank you for pointing this out. The sentence has been revised to avoid any confusion regarding its identity. Please see the revised manuscript.

Lines 403-404 “Peptide UU1 bound across the upper edge of the α2 I domain with -5.8 kcal/mol bind-ing affinity.”.

  1. Lack of evidence for the presented mechanism. It has not been demonstrated that integrin α2β1 itself stimulates further collagen I accumulation. A reference to the literature or experiments confirming such an effect would be necessary.

We appreciate the reviewer’s comment. To analyze the potential mechanism of UU1, we performed molecular docking, which suggested a possible interaction between UU1 and integrin α2β1. As mentioned before, we acknowledge that increased collagen I expression does not necessarily indicate direct binding of peptide UU1 to integrin α2β1, and other mechanisms may be involved. Our results just suggested a potential interaction. Please see the revised manuscript.

Lines 413-414 “Other integrins, such as α1β1 and α11β1 relating to bound to different type of colla-gens, might also be affected by peptide UU1. More studies need to be done to re-veal them.”

Reference:

Popov, C.; Radic, T.; Haasters, F.; Prall, W.C.; Aszodi, A.; Gullberg, D.; Schieker, M.; Docheva, D. Integrins Α2β1 and Α11β1 Regulate the Survival of Mesenchymal Stem Cells on Collagen I. Cell Death Dis. 2011, 2, e186–e186, doi:10.1038/cddis.2011.71.

  1. It was probably meant to be EGFR instead of EFPR.

We have revised it.

  1. The statement "safe and antioxidant bioactive yak bone-derived peptide" is not supported by data. No results were presented regarding the safety of UU1 or its antioxidant activity.

We thank the reviewer for the comment. Peptide UU1, derived from yak bone collagen, comes from a safe source. As noted in our previous study, it exhibited no toxicity in nematodes or the cellular assay conducted in this study, confirming its safety. Additionally, peptide UU1’s antioxidant activity was characterized in our prior study.

Reference:

Wang, Y.; Sun, Y.; Wang, X.; Wang, Y.; Liao, L.; Zhang, Y.; Fang, B.; Fu, Y. Novel Antioxidant Peptides from Yak Bones Collagen Enhanced the Capacities of Antiaging and Antioxidant in Caenorhabditis Elegans. J. Funct. Foods 2022, 89, 104933, doi:10.1016/j.jff.2022.104933.

  1. Simplified and illogical reasoning regarding the mechanisms of UU1.

We appreciate the reviewer’s comment. Molecular docking was performed to analyze the potential mechanism of UU1. Our results only suggest one potential mechanism.

  1. Lack of references to the literature or experimental evidence in some places.

We have carefully reviewed the text and added relevant citations and our experimental data to support our claims.

Round 2

Reviewer 3 Report

Comments and Suggestions for Authors

Authors have responded comprehensively and thoughtfully to the reviewer's comments. They’ve made necessary revisions to enhance the clarity and scientific rigor of the manuscript, and their responses demonstrate a proactive approach in improving their work.

Comments on the Quality of English Language

I do not assess